# Predicting Malaria Transmission Dynamics in Dangassa, Mali: A Novel Approach Using Functional Generalized Additive Models

**DOI:** 10.3390/ijerph17176339

**Published:** 2020-08-31

**Authors:** François Freddy Ateba, Manuel Febrero-Bande, Issaka Sagara, Nafomon Sogoba, Mahamoudou Touré, Daouda Sanogo, Ayouba Diarra, Andoh Magdalene Ngitah, Peter J. Winch, Jeffrey G. Shaffer, Donald J. Krogstad, Hannah C. Marker, Jean Gaudart, Seydou Doumbia

**Affiliations:** 1Malaria Research and Training Center, Faculty of Medicine, Pharmacy and Dentistry, University of Sciences, Techniques and Technologies of Bamako, Bamako BP 1805, Mali; freddy.francois.ateba@gmail.com (F.F.A.); isagara@icermali.org (I.S.); nafomon@icermali.org (N.S.); mtoure@icermali.org (M.T.); sanogodd@gmail.com (D.S.); ayouba.diarra@icermali.org (A.D.); 2Department of Mathematics, University of Quebec at Montreal (UQAM), Montréal, QC H2X 3Y7, Canada; 3Faculty of Health Sciences, University of Buea, Buea BP 63, Cameroon; magdalinebesta@yahoo.com; 4Department of Statistics, Mathematical Analysis and Optimization, University of Santiago de Compostela, Santiago de Compostela, 15782 Galicia, Spain; manuel.febrero@usc.es; 5Department of Public Health Education and Research, Faculty of Medicine and Odonto-Stomatology, University of Sciences, Techniques and Technologies of Bamako, Bamako 1805, Mali; 6Department of International Health, Johns Hopkins Bloomberg School of Public Health, Baltimore, MD 21205, USA; pwinch@jhu.edu (P.J.W.); hannah.marker@jhu.edu (H.C.M.); 7Department of Global Biostatistics and Data Science, School of Public Health and Tropical Medicine, Tulane University, 1440 Canal Street New Orleans, New Orleans, Louisiana, LA 70112, USA; jshaffer@tulane.edu (J.G.S.); krogstad@tulane.edu (D.J.K.); 8Aix Marseille University, APHM, INSERM, IRD, SESSTIM, Hop Timone, BioSTIC, Biostatistics & ICT, 13005 Marseille, France; jean.gaudart@univ-amu.fr

**Keywords:** malaria, functional model, passive case detection, meteorological indicators, Mali

## Abstract

Mali aims to reach the pre-elimination stage of malaria by the next decade. This study used functional regression models to predict the incidence of malaria as a function of past meteorological patterns to better prevent and to act proactively against impending malaria outbreaks. All data were collected over a five-year period (2012–2017) from 1400 persons who sought treatment at Dangassa’s community health center. Rainfall, temperature, humidity, and wind speed variables were collected. Functional Generalized Spectral Additive Model (FGSAM), Functional Generalized Linear Model (FGLM), and Functional Generalized Kernel Additive Model (FGKAM) were used to predict malaria incidence as a function of the pattern of meteorological indicators over a continuum of the 18 weeks preceding the week of interest. Their respective outcomes were compared in terms of predictive abilities. The results showed that (1) the highest malaria incidence rate occurred in the village 10 to 12 weeks after we observed a pattern of air humidity levels >65%, combined with two or more consecutive rain episodes and a mean wind speed <1.8 m/s; (2) among the three models, the FGLM obtained the best results in terms of prediction; and (3) FGSAM was shown to be a good compromise between FGLM and FGKAM in terms of flexibility and simplicity. The models showed that some meteorological conditions may provide a basis for detection of future outbreaks of malaria. The models developed in this paper are useful for implementing preventive strategies using past meteorological and past malaria incidence.

## 1. Introduction

In the next decade, Mali seeks to achieve the challenge of reaching the malaria pre-elimination stage [1]. The implementation of all World Health Organization (WHO) measures to prevent and control malaria remains intensive across the country [2]. According to Mali’s National Malaria Control Program (NMCP) report in 2018, among the 2,749,118 suspected cases of malaria tested, 60.28% were confirmed, of which 34.32% were children less than five years old. Compared to 2017, malaria cases increased by 16.50% in the general population, while the case fatality rate decreased from 0.76% to 0.65% [1]. 

The epidemiological situation of malaria in Mali remains highly variable, despite some improvements. Many disparities in the mortality and morbidity rates have been observed countrywide. Such observed differences in malaria burden have their explanations in both socioeconomic and environmental factors, such as education levels, occupation, use of protective measures, living standards, temperature, air humidity, rain, and wind speed [3,4,5,6,7,8]. Since malaria is a vector-borne disease, climate change can affect its transmission [9,10,11]. 

Many modelling approaches have been developed in the past, integrating climate and environmental variables to better understand the impact on malaria transmission dynamics [12,13,14,15]. Functional Data Analysis (FDA) [16,17,18] is an alternative and flexible modeling approach dealing with measurements taken over a continuum. In this case, past meteorological information (previous 18 weeks to week of interest) was considered as a function for finding the patterns that have influenced an increase in malaria cases. This novel variable selection approach [19] makes intensive use of the distance correlation [20] and is implemented in the R package fda.usc [21]. We can also notice that this new variable selection approach allows the building of more efficient models based on historical data, which fully accounts for uncertainty associated with the model selection process. 

This technique has been used recently in many modeling applications, such as influenza incidence rate modeling with climate covariates [22], but has yet to be used in the context of malaria. In this study, we use such methods to identify the underlying factors that shape the patterns of malaria prevalence in Dangassa, a rural Malian village which experiences bimodal malaria transmission dynamics [3,23]. Proper understanding of both past and future influences of environment and meteorological factors on malaria risk will help identify the relevant variables that contribute to the spread of the disease and to prevent outbreaks based on past malaria incidence rate and recognizable environmental climatic patterns. This study demonstrates that the FDA approach can be used in the malaria field with a set of recently developed functional models such as FGLM (Functional Generalized Linear Model), FGSAM (Functional Generalized Spectral Additive Models), and FGKAM (Functional Generalized Kernel Additive Models). We also describe how the results can be useful for designing targeted malaria intervention strategies. 

## 2. Materials and Methods

### 2.1. Materials

#### 2.1.1. Study Areas

This study was conducted in Dangassa (12.14 N, 8.21 W), located in Niagadina’s council, in the administrative region of Koulikoro, Mali (Figure 1). In 2012, the estimated population was 6200 inhabitants [24]. The average annual temperature and rainfall are 27.5 °C and 855 mm, respectively. The village sits at an altitude of 350 m in the Pre-Guinea savannah zone of Mali [25]. Like many villages in Mali, malaria remains a public health concern in Dangassa. The transmission dynamic is bimodal, with the start and end of the rainy season (in June–August and December–January) accounting for peak transmission. Measures such as Long Lasting Insecticidal Nets (LLINs), Artemisinin-based Combination Therapies (ACTs), and Intermittent Preventive Treatment during pregnancy (IPTp) have been in effect in Dangassa since 2008 [3,23].

#### 2.1.2. Data Source

##### Study Population and Data Collection

(1)Malaria Data

From 2012 to 2017, an observational study assessed the impact of malaria control measures at four study sites, including Dangassa [23]. An open dynamic cohort of 1400 participants of all ages (0–85 years) and sex were recruited. Passive case detection was performed at the local community health center. Free diagnosis and treatment with ACTs were provided to cohort participants with uncomplicated *Plasmodium falciparum* infection. Clinical or symptomatic malaria cases were defined as fever (temperature ≥ 37.5 °C) or history of fever in the last 48 hours, with positive rapid diagnostic test and/or positive smear by microscopy. A signed informed consent was required from randomly selected household members before they participated in the study, and parents or legal guardians gave their approval for all minors involved. Ethical approvals were obtained from the National Institutes of Health (NIAID) and from the Institutional Review Boards (IRBs) of Tulane University (FWA00002055) and the University of Sciences, Techniques and Technology of Bamako, Mali (FWA00001769) [23].
(2)Meteorological and Environmental Data

Daily and monthly meteorological data were extracted from the National Aeronautics and Space Administration (NASA) Earth Observing System Data and Information System (EOSDIS) [25] from 11 January 2012 to 31 December 2017. The data extracted for this analysis were precipitation (mm/day, 0.25°), average air temperature (°C, 0.5 × 0.625°), humidity in the ground surface (1°), and wind speed (m/s, 0.25°).

Ethical Approvals: Ethical approvals were obtained from the National Institutes of Health (NIAID) and from the IRBs of Tulane University (FWA00002055) and the University of Sciences, Techniques and Technology of Bamako in Mali (FWA00001769). Before patients were enrolled in this study in 2011 and for those enrolled after, a written informed consent was obtained from each participant or their parent/legal guardian. Please note that the cohort study protocol has been reviewed and renewed annually since that time [23].

Research Data: The dataset of malaria cases aggregated on a weekly basis are available at the level of the International Center of Excellence for Malaria Research (ICEMR) data management core (sdoumbi@icermali.org). The meteorological and environmental data are free of access, and available parts of this analysis were extracted from the EOSDIS information system (https://urs.earthdata.nasa.gov).

### 2.2. Statistical Methods

We considered several summaries (minimum, maximum, average, and amplitude) for the main covariates: temperature, humidity, wind speed, and the number of rain events. This was done to construct the set of possible candidates to be incorporated into the functional regression models: FGLM [26,27], FGSAM [28], and FGKAM [29] in all cases for predicting malaria incidence. 

The three models share the following equation:(1)$$y\left(t+1\right)=f_{1}\left(X_{1}\left(t,\ldots ,t-17\right)\right)+\cdots +f_{p}\left(X_{p}\left(t,\ldots ,t-17\right)\right)+\varepsilon$$
where y(t+1) represents the malaria incidence at a certain week, $X_{i}\left(t,\ldots ,t-17\right)$ is the whole trajectory of the covariate in the previous 18 weeks, and $f_{i}$ is a function that translates the information of the covariate to the malaria incidence and is the residual error (typically following a normal distribution).

The differences among models are based on the form of $f_{i}$:
FGLM: $f_{i}\left(X_{i}\left(t,\ldots ,t-17\right)\right)=\int_{t-17}^{t}\beta_{i}\left(u\right)X_{i}\left(u\right)du$FGSAM: $f_{i}\left(X_{i}\left(t,\ldots ,t-17\right)\right)=\sum_{k=1}^{K}f_{i}^{k}\left(\nu_{i}^{k}\right)$ with $f_{i}^{k}$ being smooth functions of $\nu_{i}^{k}$, the score of the *k*th principal component of the *i*th covariate.FGKAM: $f_{i}\left(X_{i}\left(t,\ldots ,t-17\right)\right)=f_{i}\left(X_{i}\right)$ with $f_{i}$ being a general function computed from the functional covariate using a Gaussian kernel approximation.

Except for the FGKAM, we selected a way of representing the information contained in covariate ***X***. Typically, this was done using a fixed basis like Fourier B-spline or Wavelet or using a data-driven basis like the principal components (the decomposition of the variance–covariance matrix of ***X***) or the partial least squares (the components that maximize the relationship among ***X*** and the response). 

In this paper, the functional principal components basis was chosen. This basis is quite simple to compute, and it is the one which can explain more of ***X*** with fewer elements. For designing the covariates that integrate our functional models, for selecting relevant information, and for avoiding variates with high collinearities, we used the distance correlation measure [20]. The primary advantage over its competitors is that it portrays independence among two covariates, no matter the distribution or the dimension of the covariates. 

The distance correlation takes value in the interval [0, 1], where zero indicates complete independence and one indicates full dependence. Its scale is like that of the coefficient of determination, although the distance correlation has no such simple interpretation in terms of the explained variability of the response. In any case, higher values of the distance correlation mean higher dependence, and the same authors that proposed the distance correlation have proposed an independence test for distance correlation [21]. The selection of the covariates was done using the algorithm described in the novel variable selection approach [19], which seems to select non-sparse models. To assess the performance of the algorithms related to the previous models in our data set, we have proposed a comparison based on some common characteristics of those models, such as the R-sq. (adj), Mean Squared Prediction Error (MSPE), and predictive ability. The best choice is measured in terms of prediction coverage (predictive ability, %), checking the real coverage of the 95% prediction intervals for data not included in the estimation process.

All statistical analyses was performed by using the R 4.0.0 (R Foundation for Statistical Computing, Vienna, Austria) [30] software and the Rpackage fda.usc [21], where the methods for variable selection and the functional regression models are implemented. ArcGIS 10.3 (ESRI, Redlands, CA, USA) [31] has been used for cartography of our study site.

## 3. Result

### 3.1. Descriptive Analysis of the Functional Data

In Dangassa, based on the data collected in the period 2012–2017, the minimum number of cases was three per 1000 person-weeks when the maximum number of cases was 70 per 1000 person-weeks. Descriptive analyses of each of the functional covariates (Figure 2) were done by comparing the malaria incidence in four groups, an intuition born from the quantile distribution observed on the data set descriptive analysis against the pattern of the measurements over the previous 18 weeks: low, medium low, medium high, and high. For each group, we have computed the average of the curves that lead to the response group as a way of indicating past patterns of the curves which lead to higher or lower rates. This simple descriptive analysis revealed that the mean humidity pattern of over 65% produced the highest incidence rate while, below 50%, it produced the lowest incidence rate. A mean rain event pattern of more than two rain events produced the highest incidence rate, and below that threshold, it produced the lowest incidence rate. The peaks of humidity and rain events happen about 12–13 weeks before the highest malaria incidence rate. A similar phenomenon occurs with temperature. A mean temperature pattern below 27 °C led to a high incidence rate, particularly with a deep valley under 26 °C, 10 weeks in advance to peak incidence rates. Consecutive weeks with temperature over 28 °C led to the lowest incidence rates. A mean wind speed pattern below 1.8 m/s for at least 10 weeks led to a high incidence rate. Altogether, the descriptive analysis points out that if we observe an episode with high humidity, a high number of rain events, and low wind speed about 10–12 weeks before, we will likely witness a malaria outbreak. The effect of past incidence pattern is less interesting (see Figure 3), but we observed a slow decline for lower incidence rates and a high increasing pattern at 10 weeks.

As a second part of this descriptive analysis, we compute the importance of each functional covariate, taking values in the interval [*n*−17, *n*] with the response evaluated in *n* + 1 and *n* + 2 to find out if the information chosen has relevance for predicting the malaria incidence. Given the different nature of the variates (some functional and some scalar), the only choice is the distance correlation proposed by Székely et al. [20]. The distance correlation among response and functional covariates are provided in Table 1. In order of relevance, fHumidity has the highest value (0.404 and 0.420), and then fRainNb (0.363 and 0.390) has the next highest, closely followed by fWindspeed (0.357 and 0.350) and finally fTemperature (0.267 and 0.240) and fIncidence (0.256 and 0.220). The relationship among the past values of incidence rate (fIncidence) with its future suggest that there is no strong temporal dependence in the malaria incidence rate. 

The relatively high values among covariates (Table 2) suggest a great interdependence among them. This interdependence must be considered in the construction of the regression models. If not, the inclusion of some covariates may interact with others to hide relevant effects.

### 3.2. Constructing a Functional Regression Model (FGSAM) for Malaria Incidence Rate

Using the information from the previous 18 weeks of the same functional covariates for predicting the malaria incidence, we have constructed a FGSAM model. We have also tried other weekly summaries (amplitude, minimum, or maximum) but with no better success. To construct the FGSAM model, all covariates were represented by their first three principal components. The algorithm described [19] was applied to select the final covariates in the FGSAM model that obtained an adjusted R-sq (0.673) and deviance explained (72.4%), identifying four pertinent partial functions (six if we extend the confidence to 90%). The results can be seen in Table 3, where the order of the rows reflects the pertinence of each covariate. The second covariate (fWindspeed) was selected, although fRainNb had a higher distance correlation than fWindspeed (Table 2). The relatively high interdependence between fRainNb and fWindspeed perhaps influences the *p*-value column that accounts for the pertinence of each row. The covariate fTemperature was not selected in the model, meaning that, given the other covariates, there is nothing new that this covariate can add. The column edf (estimated degrees of freedom) shows the complexity of the information provided by the particular component. 

The FGSAM model is quite flexible and powerful, but its interpretation is not easy, as shown in Table 3. Every row of Table 3 is the combination of a principal component (that must be interpreted itself) jointly with a smooth function on the scores of that principal component. Therefore, to interpret the contribution of each covariate, we must combine both interpretations. Figure 4 shows the chosen principal component (PC) (left column) and its associated function (right column). The first row of Figure 4 corresponds to the effect of fHumidity and the first PC and can be interpreted in terms of its difference with respect to the zero line, which represents the average humidity in the previous 18 weeks. Therefore, PC1 of fHumidity represents the level of fHumidity with respect to its mean. Positive scores of PC1 represent curves of fHumidity constantly over the mean in the last 18 weeks, and negative scores represent curves of fHumidity constantly below the mean. The scores are represented in the right column or on the *x*-axis. 

The shape of the function with respect to these scores means that positive scores (curve of fHumidity above the mean) lead to fewer cases of malaria (the function for positive values is below zero). Negative scores (curve of fHumidity below the mean) lead to an increased malaria incidence (function slightly over zero baseline). 

The interpretation for the second row (PC1 of fRainNb) is similar, but on the contrary, positive scores (number of rain events in the last 18 weeks over the mean) lead to an increase in malaria incidence whereas negative scores (number of rain events below the mean) slightly decrease the malaria incidence (smooth function is below the zero line). The third row corresponds to PC2 of fRainNb, and its shape corresponds to curves that are below the mean before week −8 and over the mean after that. Therefore, positive scores (curves with that shape) now slightly decrease the incidence and negative scores (curves over the mean in the interval [−17, −8] and below the mean in [−8, 0]) increase the incidence. Indeed, the shape of that function suggests that the relationship among PC2 and the response is linear. The rest of the rows can be interpreted in the same way, although close proximity of the smooth function to the zero line suggests weak effects of these covariates (fWindspeed and fIncidence) on malaria incidence. 

### 3.3. Comparing Different Functional Models for Dangassa Data

We have built three models based on the pertinence of covariate curves of the previous 18 weeks of malaria incidence rate and on meteorological and environmental data observed in Dangassa using the same algorithm described in the novel variable selection approach [19]. The set of relevant covariates differs from one type of model to another. For instance, when using FGLM, all covariates were included, obtaining adjusted R-sq (0.579) and deviance explained (61.20%) as its best result. The three models were compared in terms of their predictive coverage. 

The last 40 weeks of data were used as a validation sample to check the predictive performance of the three models: for every week in the validation sample, estimation of the models was done with the past data and prediction levels of incidence along a predictive interval at the 95% level for that estimation. The prediction level was used for computing the MSPE in the usual way. The predictive coverage was estimated, counting how many times the true future value was inside the prediction interval. The results are summarized in Table 4. Although in terms of adjusted R-sq FGSAM provides the highest value, the best predictive coverage (closer to the nominal level) is provided by FGLM.

The predictive estimation and their 95% predictive interval are plotted in Figure 5, confirming that the FGKAM model has the best predictive abilities. It seems that FGLM and FGSAM provide excessively optimistic prediction intervals (see the amplitude of the intervals), derived by an underestimation of the predictive variance. Unfortunately, the interpretation of FGKAM is not an easy task because there are no simple tests that could help. From a summary of the model, it is possible to point out that the fRainNb, fHumidity, and fWindspeed are the most relevant factors related to malaria incidence but that it is not possible to derive simple rules relating the covariates and the response. The past fIncidence and fTemperature have a clearly lower influence in the response, although it is not negligible. 

Our findings show that the role of temperature on malaria dynamics is complicated and indicate an indirect but ignored impact of air temperature on the increase of malaria transmission through reduction of larval habitats and vector density. Once more, FGSAM has been shown to be a good compromise among flexibility and simplicity of interpretation, but FGLM provides the best predictive results. The results obtained by FGSAM confirm that this prediction problem is not linear, i.e., the effect of the covariates on the response is more complicated than linear models.

## 4. Discussion 

In this paper, we have proposed the use of a functional approach to predict malaria outbreaks based on a rigorous selection of the covariates that contribute the most to the spread of malaria in Dangassa. The use of the distance correlation allowed us to identify, as shown extensively in the literature, some environmental variables that influence malaria incidence [3,6,7,8,12]. 

In Dangassa, humidity, number of rain events, wind speed, temperature, and past incidence were the climate covariates associated with malaria incidence, but here, we have been able to determine an order of their relevance in influencing malaria transmission dynamics. The functional models used (FGLM, FGSAM, and FGKAM) allowed us to use past malaria incidence as an additional covariate, although its contribution to understanding malaria transmission dynamics and predicting future malaria outbreaks in Dangassa is small (quite negligible). With the functional modeling approach, we have taken time as a continuum and used curves rather than point-time estimations, as is done in many other approaches [13]. 

To predict malaria outbreaks, our results suggest we pay attention to a particular meteorological configuration: humidity greater than 65%, more than two rain events, and wind speed levels < 1.8 m·s^−1^. If we detect this particular configuration, then outbreak preparedness should start, as it means that a malaria outbreak will probably occur in Dangassa 10–12 weeks later. Using these functional model predictions, our aim is to try to prevent outbreaks and to raise alerts in order to prepare both public health authorities and populations in advance of an outbreak.

It has been shown in many studies that malaria-to-malaria transmission dynamics depend not only on a few meteorological conditions like humidity, rain, wind speed, vegetation, and air temperature at the ground level but also on factors such as sociocultural behaviors, access to health care, level of education [32,33], and use of vector protection tools [6,9]. One limitation of our functional models is that they include only climate covariates. Our results show that FGKAM, FGLM, and FGSAM obtained explained deviances of 75.10%, 61.20%, and 72.40%, respectively’ however, we could improve the explained deviance of our functional models by including non-climate covariates related to malaria transmission dynamics, including data from the vectors such as biting rate and genetic resistance mechanism to anti-vectoral drugs [11,34]. It could be beneficial to add variables related to the implementation of preventive measures like Seasonal Malaria Chemoprevention (SMC) and long-lasting insecticide-impregnated nets (LLINs) in our functional models, as they can deal with scalar covariates in predicting the response curves.

During the project of West African International Center of Excellence for Malaria Research (ICEMR-1) at the base of this study, the SMC effect on malaria indicators in children under five years old living in Dangassa was investigated. A monthly curative dose of SP + AQ (sulfadoxine-pyrimethamine + amodiaquine) was given to each child during malaria transmission season (August to October). A significant reduction in both malaria incidence and gametocyte prevalence levels in children under five years due to the SMC treatment was found [35]. This has surely shaped malaria transmission dynamics in a particular way, as our models could predict a greater number of cases during a period where SMC was not being distributed and a lower number of cases in an SMC implementation period. If such information is included in our models, we could improve their explained deviance and their accuracy.

In Dangassa, malaria control strategies rely on the use of long-lasting insecticide-impregnated nets (LLINs), ACT for treatment, and sulfadoxine-pyrimethamine (SP) for intermittent preventive treatment of pregnant women (IPTp). No artemisinin resistance genetic background was found [36]. In the context of Dangassa, ACTs and SP resistance did not contribute to malaria recrudescence and, consequently, incidence. Our results could not suffer from the non-inclusion of that reality in the functional modeling approach.

In this study, we have explored three different types of functional models in the field of malaria: FGLM (Functional Generalized Linear Model), FGSAM (Functional Generalized Spectral Additive Models), and FGKAM (Functional Generalized Kernel Additive Models). It has been necessary to compare them in the case of malaria because some specific differences in the process of parameter estimations could have favored one. In fact, FGLMs are the extension of classical GLMs, used as functional predictors, and simply consists of replacing the linear combination of the covariates by the inner product in the functional space [26]. This could be a limitation in certain situations, where a functional datum could contain different information depending on the semi-metric used. However, we have not reached the limitations of this models due to the nature of our data. In the case of FGSAM, the estimation of the partial functions is made through the functional principal component (FPC) scores. This model makes use of spectral decomposition of the covariance operator of the matrix of covariates X, although the use of other basis representations is possible or even, in certain cases, desirable. The GSAM model has an increasing flexibility while avoiding the curse of dimensionality. Indeed, the fact that the FPC scores are always uncorrelated for every functional covariate ensures that the estimation of partial functions associated with that covariate will not suffer concurvity problems (some smooth terms could be approximated by one or more of the other smooth terms) [27,28]. The fact that the FGSAM model does not suffer concurvity problems makes it of potentially appropriate use in our current situation in the field of malaria. The last model we used here is FGKAM, which is based on a mixture of the Iteratively Reweighted Least Squares (IRLS) and Backfitting algorithms adapted to the functional context. It allows the nonparametric estimation of partial functions [29].

Many models developed in the field of malaria do not order the covariates by order of relevance. They do not evaluate the strength of the signal coming from the covariates on the response (malaria incidence for example); our functional models handled that issue. For instance, in the work carried out by Ateba and al. [3] in the same site of Dangassa, temperature was included in our Generalized Additive Models(GAMs) models, but it was not possible to quantify to what extent temperature contributed to explaining malaria incidence. Here, in our framework, we have been able to determine the order of relevance of the factors (mostly meteorological) that have been included. It has been made clear in the context of Dangassa that some factors like past incidence and temperature contribute less (almost not at all) in explaining the increase in malaria incidence in Dangassa. As for the other factors, humidity, windspeed. and rain contributed the most, in that order (based on the distance correlation [20]). 

This study has proposed three functional models, although none of them were a clear winner. A compromise between predicting abilities and ease in interpreting results is needed when choosing a model to make a prediction of malaria incidence. 

The added value of using functional modeling here has been to clearly identify a particular pattern of meteorological conditions that may occur in Dangassa 10 to 12 weeks before observing malaria outbreaks. This finding indicates that both public health authorities and meteorological offices can assist in decision making to reduce the burden of the disease by raising alerts when particular patterns of meteorological conditions arise. 

Our results did not provide us with a clear decision on which of the models we should applied in the prediction of malaria outbreak. Ee should always be cautious and put into context our results and their interpretations as to comprise between model flexibility and simplicity in the interpretation of results.

## 5. Conclusions

A geo-epidemiological approach using functional models can be extremely useful to health managers in allocating resources in advance for epidemic outbreak control and management. The National Meteorological Agency of Mali could play a key role in malaria outbreak prevention and preparedness by raising alerts if particular meteorological patterns occur. 

## Figures and Tables

**Figure 1 ijerph-17-06339-f001:**
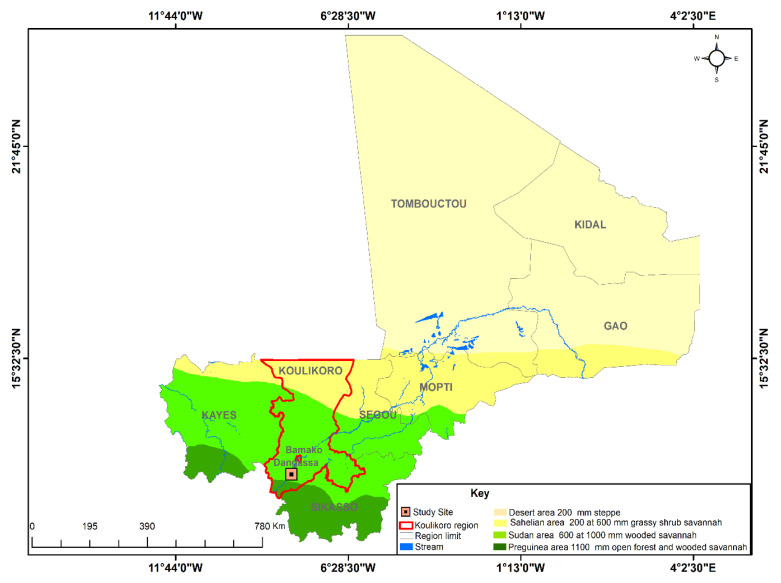
Map of Mali indicating the location of Dangassa ^a. a^ The study site is indicated by a black point. The map has been made based on the cartography of Mali, the mean normalized difference vegetation index (NDVI) reported was downloaded as a raster from NASA Giovanni for the time period 11/01/2012–31/12/2017. Source: ICER Mali/MRTC-OKD/DEAP/GIS Unit, 2020.

**Figure 2 ijerph-17-06339-f002:**
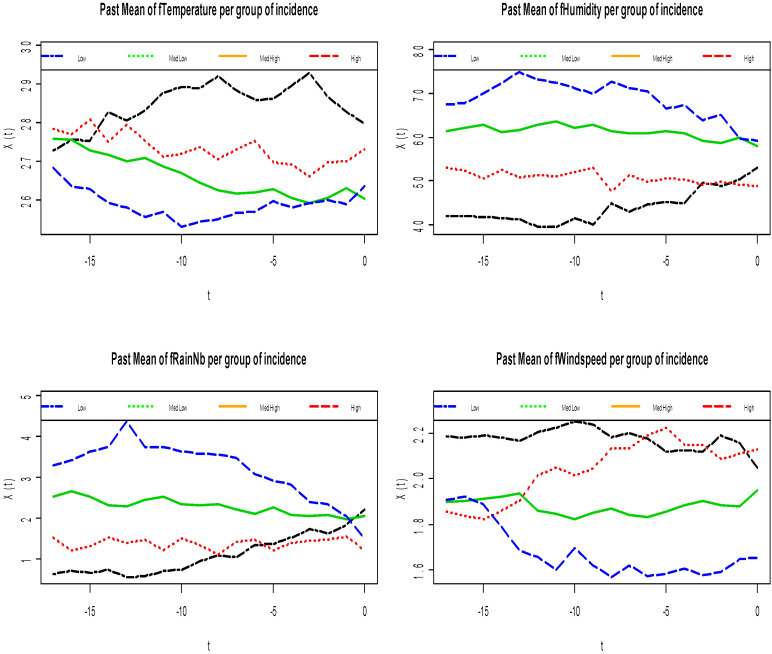
Mean curves of past functional meteorological covariates per group of malaria incidence ^b^. ^b^ Clockwise from top left: fTemperature, fHumidity, fWindspeed, and fRain mean curves groups. The groups are constructed based on values of the quantile of the incidence values: low [0, 2.5] (dark blue line), medium low (2.5, 5] (green line), medium high (5, 10] (black line), and high (10, 20] (dark red line). Altogether, if we observe a mean humidity pattern (>65%), mean rain events (>2), and mean wind speed (<1.8 m.s^−1^) about 10–12 weeks before, then we will probably suffer a peak in malaria incidence in Dangassa.

**Figure 3 ijerph-17-06339-f003:**
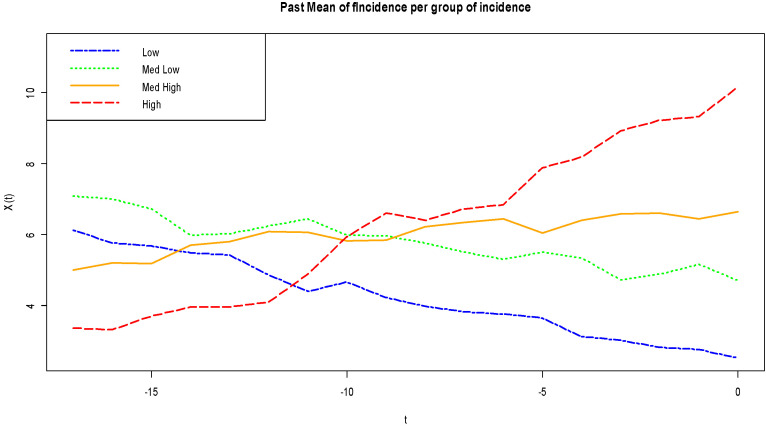
Mean curve patterns of past malaria incidence per group of incidence ^c. c^ The groups are constructed based on values of the quantile of the incidence values: low [0, 2.5] (dark blue line), medium low (2.5, 5] (green line), medium high (5, 10] (black line), and high (10, 20] (dark red line). Descriptive analysis shows a slow decline for lower incidence rates and a high and abrupt increasing pattern at weeks 10.

**Figure 4 ijerph-17-06339-f004:**
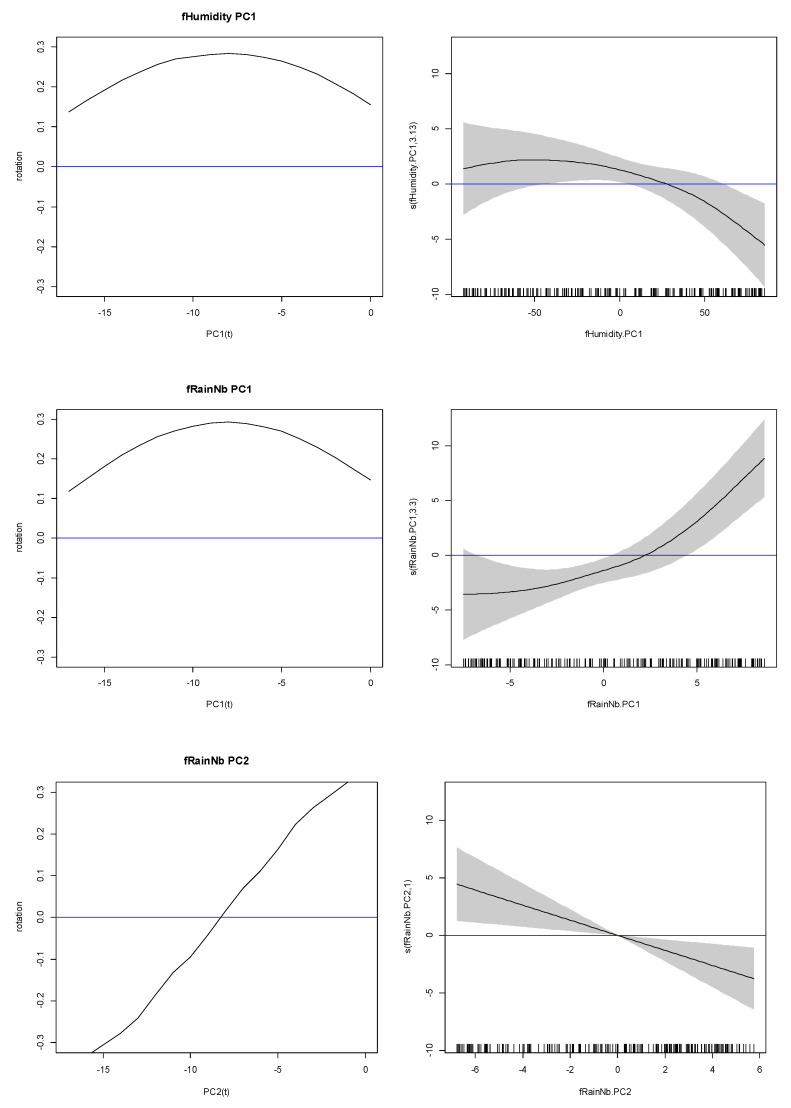
Some chosen principal component (PC) and their associated smooth functions ^g^. ^g^ The shape of fHumidity.PC1 over the mean leads to fewer cases of malaria, and the curve of fHumidity below the mean leads to an increased malaria incidence. The shape of fRainNb.PC1 over the mean leads to an increase in malaria incidence, whereas the number of rain events below the mean slightly decreases the malaria incidence. The relationship among fRainNb.PC2 and the response is linear. The effect covariates fWindspeed and fIncidence on malaria incidence are weak.

**Figure 5 ijerph-17-06339-f005:**
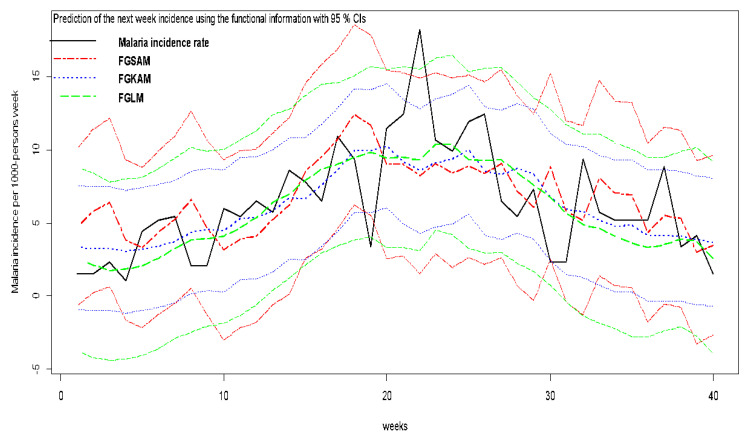
Prediction of the raw rates (cases per 1000/pop) in the village of Dangassa ^i^. ^i^ The predictions have been made in the validation period for the Functional Generalized Spectral Additive Model (FGSAM), the Functional Generalized Linear Model (FGLM), and the Functional Generalized Kernel Additive Model (FGKAM). We used functional information based on past incidence and meteorological covariates with 95% Confidence Intervals (CIs). The black solid line represents the validation set (40-week)-based incidence (observed). The two dashed red lines represent the FGSAM predictions and its 95% CI. The dotted blue lines represent the FKAM predictions and its 95% CI. The green long dash lines represent the FGLM predictions and its 95% CI. All 3 models performed well, but FGLM has 95% CI curves closer to the validation set incidence curves. FGLM seems to have the best tuned 95% CI prediction bandwidth.

**Table 1 ijerph-17-06339-t001:** The distance correlation among the functional covariates and the response at week *n* + 1 and *n* + 2 ^d^.

Functional Covariates	Incidence (*n* + 1)	Incidence (*n* + 2)
fIncidence (*n* − 17, …, *n*)	0.256	0.220
fWindspeed (*n* − 17, …, *n*)	0.357	0.350
fRainNb (*n* − 17, …, *n*)	0.363	0.390
fTemperature (*n* − 17, …, *n*)	0.267	0.240
fHumidity (*n* − 17, …, *n*)	0.404	0.420

^d^ In Dangassa, malaria incidence is influenced in order of relevance by fHumidity (0.404 and 0.420), by fRainNb (0.363 and 0.390), by fWindspeed (0.357 and 0.350), and finally by fTemperature (0.267 and 0.240) and fIncidence (0.256 and 0.220). We discovered that there is no strong temporal dependence in the malaria incidence rate.

**Table 2 ijerph-17-06339-t002:** Distance correlation among functional covariates ^e^.

Functional Covariates	fIncidence	fTemperature	fHumidity	fRainNb	fWindspeed
**fIncidence**	1.000	0.457	0.556	0.604	0.585
**fTemperature**	0.457	1.000	0.519	0.430	0.387
**fHumidity**	0.556	0.519	1.000	0.887	0.705
**fRainNb**	0.604	0.430	0.887	1.000	0.691
**fWindspeed**	0.585	0.387	0.705	0.691	1.000

^e^ The dependence among functional variates is measured by the value of the correlation of distances. The relatively high values among all the covariates suggests a great interdependence among them. fHumidity and fRain have the strongest correlations (0.887), while fTemperature and fWindspeed have the lowest correlations.

**Table 3 ijerph-17-06339-t003:** Pertinence of partial candidate smooth functions to enter into the Functional Generalized Spectral Additive Model (FGSAM) and the nature of their relationship to the response: pertinent curves to enter in the FGSAM model ^f^.

Curves	Edf	Ref.df	F	*p*-Value
s(fHumidity.PC1)	3.126	4.003	3.865	0.005
s(fWindspeed.PC2)	2.000	2.536	2.259	0.075
s(fRainNb.PC1)	3.304	4.199	9.457	<0.001
s(fRainNb.PC2)	1.000	1.000	7.840	0.006
s(fIncidence.PC1)	8.544	8.910	4.551	<0.001
s(fIncidence.PC3)	1.000	1.000	2.885	0.091

^f^ fHumidity, fWindspeed, and fRainNb are in that order the most important candidate smooth curves to enter into the FGSAM model. The covariate fTemperature was not selected in the model. The information provided by the fHumidity.PC1, fWindspeed.PC2, fRainNb.PC1, and fIncidence.PC1 components to the response is quite not linear (complex).

**Table 4 ijerph-17-06339-t004:** Comparison of the predictive abilities of the functional models Functional Generalized Linear Model (FGLM), Functional Generalized Spectral Additive Model (FGSAM), and Functional Generalized Kernel Additive Model (FGKAM) ^h^.

Goodness-of-Fit Measures of the Functional Models	FGKAM	FGLM	FGSAM
Adjusted R-sq (%)	65.70	57.90	67.30
Dev. Explained (%)	75.10	61.20	72.40
MSPE	7.52	7.50	11.38
Pred. coverage (%)	90.00	95.00	92.50

^h^ Here, we display some goodness-of-fit measures R-sq(adj), Mean Square Prediction Error (MSPE), and predictive coverage as a tool to compare the functional models FGLM, GGSAM, and FGKAM. In terms of the predictive abilities, all models performed well, none did better than the others. FGSAM fit the best with adjusted R-sq (67.3%), but FGLM had the best predictive coverage (95%) and FGSAM obtained the best explained deviance (75.1%).

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
