# Peer review of "Predicting Malaria Transmission Dynamics in Dangassa, Mali: A Novel Approach Using Functional Generalized Additive Models"

_ijerph, 2020, doi:10.3390/ijerph17176339_

Round 1

Reviewer 1 Report

  • The abstract is clearly and accurately describing the content of the article submitted.
  • Mali aims to reach the pre-elimination stage of malaria by the next decade. The manuscript emphasizes the importance of studies using functional regression models to predict the incidence of malaria as a function of past meteorological patterns to better prevent, and to act proactively against impending malaria outbreaks. Furthermore, the article also discusses the perspective created by our increasing understanding of the modeling approaches and how it will be able to improve preventive strategies using past meteorological and past malaria incidence.
  • The methods described are comprehensible. However, it will be recommended that more methodological details be provided regarding the asymptomatic cases (if have) and about Seasonal Malaria Chemoprevention (SMC), implemented since 2012. It is important to comment about ACTs and SP resistance, if this contributed to malaria recrudescence and consequently incidence.
  • The number of tables and figures are adequate, and the data is informative.
  • Concerning the discussion, it is suggested to better explain the functional models used (FGLM, FGSAM, and FGKAM). It would be important to comment more on this finding to make a predictions of malaria incidence. Moreover is recommended to discuss the SMC and no SMC in the period 2012 to 2017.

Author Response

Thanks a lot for such a great contribution to our work.

Thanks once again.

Reviewer 2 Report

The manuscript by Ateba et al. describes the analyses of regression models used to predict the incidence of malaria at a community health center as a function of prior meteorological patterns. Three functional models were explored. Patterns of air humidity levels (>65%), two or more consecutive rain episodes and mean wind speeds (<1.8 m/s) were predictive. Among 3 the 3 models explored, the functional generalized linear model was the most predictive, however a functional generalized spectral additive model shows value in terms of application.

The manuscript is well written (see below for some minor suggestions).  The methods appear sound and data are well presented. The discussion and conclusion are appropriate.

Would like to see more details re: IRB approval/review as appropriate.

Some minor issues.

Line 43: change “consecutives” to “consecutive”

Line 70: space after [19] (Here and throughout)

Line 72: Consider rewording the beginning of the sentence for clarity

Lines 113/161: Any statement of IRB review of consent document or approval of study by the participating institutions?

Lines 182/183: Consider rewording re: “probably suffer a peak”

Line 336: space between “95%” and “CI” (see throughout for consistency)

Author Response

Thanks a lot for contributing so much in our work.

Thanks once again.

Reviewer 3 Report

The work presented by Ateba and colleagues is of interest in regard of the current aim to reach malaria pre-elimination. Indeed, beyond the classical strategies deployed to fight malaria, it is well known that additional tools are needed to reach and to maintain the level of the disease pre-elimination. In this line, prediction tools can help as an early warning system to address the risk of malaria transmission.

Minor remarks

The model integrates malaria transmission data classified according to different levels, however, the authors do not give any indication on the values of disease incidence corresponding to the high or low level, for example.

Also, give the indication about the level of malaria transmission in Dangassa. The author just indicate that malaria burden is huge. Authors must be clear on the malaria transmission pattern in Dangassa: in line 93-94 it is stated that transmission occurs through the year, but in line 72-73 the transmission is seasonal

The discussion section deserve to be developed a little more and take into account the limits of their model and put it in context, i.e. there are several factors that contribute to malaria transmission such as access to health care, use of vector protection tools. It is true that the model is exclusively based on climate data, do  the authors tried to see if there are other confounding effects? Indeed the period of the study is relatively long, it is not possible that for example the absence or non-use of mosquito nets is the main factor explaining the high incidence. At least these aspects should be addressed in the discussion.

The figures 2 to 4 had very bad resolution and the legend is unreadable

Author Response

Thanks a lot for your contribution to our work.

So happy to read and answer to your questions. 

Round 2

Reviewer 1 Report

The authors adequately answered all questions and improved the text. The manuscript is now ready to be published.